# Risk factors for neuropsychiatric symptoms in patients with Parkinson's disease during COVID-19 pandemic in Japan

**Fukiko Kitani-Morii**[1,2], **Takashi Kasai**[2]*, **Go Horiguchi**[3], **Satoshi Teramukai**[3], **Takuma Ohmichi**[2], **Makiko Shinomoto**[2], **Yuzo Fujino**[2], **Toshiki Mizuno**[2]

**1** Department of Molecular Pathobiology of Brain Diseases, Kyoto Prefectural University of Medicine, Kyoto, Japan, **2** Department of Neurology, Kyoto Prefectural University of Medicine, Kyoto, Japan, **3** Division of Data Science, The Clinical and Translational Research Center, University Hospital, Kyoto Prefectural University of Medicine, Kyoto, Japan

* kasaita@koto.kpu-m.ac.jp

## Abstract

The worsening of neuropsychiatric symptoms such as depression, anxiety, and insomnia in patients with Parkinson's disease (PD) has been a concern during the COVID-19 pandemic, because most people worked in self-isolation for fear of infection. We aimed to clarify the impact of social restrictions imposed due to the COVID-19 pandemic on neuropsychiatric symptoms in PD patients and to identify risk factors associated with these symptoms. A cross-sectional, hospital-based survey was conducted from April 22, 2020 to May 15, 2020. PD patients and their family members were asked to complete paper-based questionnaires about neuropsychiatric symptoms by mail. PD patients were evaluated for motor symptoms using MDS-UPDRS part 2 by telephone interview. A total of 71 responders (39 PD patients and 32 controls) completed the study. Although there was no difference in the age distribution, the rate of females was significantly lower in PD patients (35%) than controls (84%) (P < 0.001). Participants with clinical depression (PHQ-9 score ≥ 10) were more common in PD patients (39%) than controls (6%) (P = 0.002). Multivariate logistic regression analysis revealed that an MDS-UPDRS part 2 score was correlated with the presence of clinical depression (PHQ-9 score ≥ 10) and clinical anxiety (GAD-7 score ≥ 7) (clinical depression: OR, 1.31; 95% CI, 1.04–1.66; P = 0.025; clinical anxiety: OR, 1.36; 95% CI, 1.07–1.72; P = 0.013). In the presence of social restrictions, more attention needs to be paid to the neuropsychiatric complications of PD patients, especially those with more severe motor symptoms.

## Introduction

The pandemic of coronavirus disease 2019 (COVID-19) and subsequent state of emergency forced people to focus on the infection and lower the priority of care for chronic diseases. Older people, who often have underlying medical conditions, have an increased mental burden, because they were reported to be at a higher risk for severe disease [1]. While Japan's state

**Data Availability Statement:** All relevant data are within the manuscript and its Supporting Information files.

**Funding:** This work was supported by Grants-in-Aid (18K07506 to T.K.; 20K16605 to T.O.) from the Ministry of Education, Culture, Sports, Science and Technology (MEXT) of Japan.

**Competing interests:** The authors have declared that no competing interests exist.

of emergency did not introduce legal penalties for leaving the house, there was concern that self-isolation and social distancing, which were strongly advocated, would worsen physical inactivity and mental instability, especially in the elderly [2, 3].

Parkinson's disease (PD) is one of the most frequent neurodegenerative diseases that is predominant in the elderly. In addition to motor symptoms like bradykinesia, rigidity, and tremor, non-motor symptoms such as depression, anxiety, and sleep disturbance occur from the early to advanced stages of PD [4–6]. There was concern that the increased mental burden and restrictions placed on exercise due to the COVID-19 pandemic negatively affect both motor and non-motor symptoms [7]. Indeed, several groups from around the world have reported higher rates of neuropsychiatric problems in PD patients due to the social restrictions imposed following the surge of infections [8–13]. Notably, a report from Netherland clearly showed that PD patients with more COVID-19 related stressors had more PD symptoms through increased mental stress [14]. However, we need to accumulate more cases from diverse regions to more closely examine the impact of the COVID-19 pandemic on PD patients, because there are regional differences in the severity of the infection and social restrictions.

The purpose of this study was to assess the severity of depression, anxiety, and insomnia in PD patients in Japan experiencing social stresses caused by the COVID-19 pandemic, and to identify factors associated with severe non-motor features and subjective worsening of motor and non-motor symptoms.

## Materials and methods

### Study design and participants

The study protocol was reviewed and approved by the institutional ethics review boards of Kyoto Prefectural University of Medicine in accordance with the Helsinki Declaration (ERB-G-12). Written informed consent was provided by all survey participants. No minors were among the participants. According to a priori power analysis (with 95% power and 5% type I error rate) of the results of previous studies [8–10], the minimum number for the sample was found to be 41 patients for the comparative study of the prevalence of psychiatric symptoms during COVID-19 pandemic in PD and control groups. This study was a cross-sectional, single hospital-based survey conducted from April 22, 2020 to May 15, 2020. Japan's state of emergency was imposed from April 7, 2020 to May 22, 2020. PD patients who regularly visited the outpatient clinic of Kyoto Prefectural University of Medicine were asked to participate in this study. The family members of each PD patient were recruited to the study as controls. Participants were asked to complete paper-based questionnaires by mail and respond to a telephone interview.

### Outcomes measures and patients' characteristics

We assessed symptoms of depression, anxiety, and insomnia both in PD patients and controls using the Japanese version of questionnaires such as the 9-item Patient Health Questionnaire (PHQ-9), 7-item Generalized Anxiety Disorder (GAD-7), and 7-item Insomnia Severity Index (ISI), respectively [15–18]. The scores for each assessment method were classified into the following categories: PHQ-9, normal (0–4), mild (5–9), moderate (10–14), moderate to severe (15–19), and severe (20–27) depression; GAD-7, normal (0–4), mild (5–9), moderate (10–14), and severe (15–21) anxiety; ISI, normal (0–7), subthreshold (8–14), moderate (15–21), and severe (22–28) insomnia (15–18). The cutoff values for PHQ-9, GAD-7, and ISI were 10, 7, and 15, respectively. Participants with scores higher than the cutoff were considered to have "clinical depression", "clinical anxiety", and "clinical insomnia". In addition, all participants

were asked if they had experienced a subjective worsening of motor performance, anxiety, and insomnia.

PD patients were further evaluated based on the Movement Disorder Society Unified Parkinson's Disease Rating Scale (MDS-UPDRS) part 2, which assesses the patient's subjective motor experiences of daily living [19]. The questions and possible responses were read out over the phone, and the patient responded accordingly. The following information was obtained from medical records of PD patients: sex, age, whether they lived alone, duration of PD, Hoehn & Yahr (HY) stage, non-motor symptoms (such as cognitive impairment, hallucinations, and rapid eye movement sleep behavior disorder [RBD]), medications such as L-DOPA, dopamine agonist, psychiatric medicines (including selective serotonin reuptake inhibitor, serotonin noradrenaline reuptake inhibitor, and tricyclic/tetracyclic antidepressant, atypical antipsychotics), and sleeping pills. Controls provided the following information: sex, age, and whether they lived alone.

## Statistical analysis

PHQ-9, GAD-7, and ISI scores are expressed as the median and interquartile (IQR) because they do not show a normal distribution. A Mann-Whitney U test was adopted to compare each questionnaire score between PD patients and controls. For PD patients, uni- and multivariate logistic regression analyses were performed to determine potential risk factors for clinical depression, anxiety, and insomnia, and subjective worsening of motor and non-motor symptoms. The relationship between risk factors (sex, age [< 70, 70–79, ≥ 80], duration of PD [years, < 5 or ≥ 5], HY stage [0–2 or ≥ 3], scores of MDS-UPDRS part 2 [Note: this scale was treated as continuous variables in the analysis], L-DOPA dose [mg, < 600 or ≥ 600], and use of dopamine agonist) and outcomes are expressed as odds ratios (ORs) and 95% confidence intervals (CIs). Univariate logistic regression analysis was performed for all variables. For the multivariate model, the L-DOPA dose and use of dopamine agonists were adopted as variables if their P-value was greater than 0.2 on univariate analysis in addition to basic characteristics (including sex, age, duration of PD, HY stage, and scores of MDS-UPDRS part 2). Data analysis was performed using SPSS version 26 (IBM, Armonk, NY, USA) and SAS version 9.4 (SAS Institute, Inc., Cary, NC, USA) under the supervision of statisticians. A P-value of less than 0.05 indicated significance, and all reported P-values were 2-sided.

## Results

### Participants' characteristics

A total of 88 patients and their family members (44 PD patients and 44 controls) were invited to participate in the study, and 71 (80%) responders completed the survey. Of the 71 participants, 39 (54%) were PD patients and 32 (45%) were controls. The response rates for PD patients and controls were 88 and 72%, respectively. The rate of females was significantly lower in PD patients (PD patients vs. controls: 14 [35%] vs. 27 [84%], P < 0.001). There was no significant difference in age between PD patients and controls (mean age ± standard deviation of PD patients vs. controls: 72.3 ± 10.9 vs. 66.4 ± 13.8, P = 0.058) and most patients in both groups were 70–79 years of age (PD patients vs. controls: 18 [46%] vs. 15 [46%], P = 0.078). Only three (4%) participants were living alone. In 39 PD patients, 17 (43%) had a disease duration of 5 years or more, 27 (69%) were classified into HY stage 3 or more, and the median MDS-UPDRS part 2 score was 17 (interquartile range: 10.0 to 20.5). The rates of cognitive impairment, hallucinations, and RBD were 6 (15%), 4 (10%), and 6 (15%), respectively. There were 9 (23%) patients who were prescribed 600 mg or more of L-DOPA, and dopamine agonists were taken by 15 (38%) patients. Psychiatric and sleeping medicines were used in 8

(20%) and 13 (33%) patients, respectively (Table 1). Among PD patients, there was no difference between female and male participants in disease duration, HY stage, non-motor symptoms, therapeutic medications, or MDS-UPDRS part 2 scores (S1 Table).

## Severity and scores of measurements

Compared with controls, significantly more PD patients presented with clinical depression using PHQ-9 (PD patients vs. controls: 15 [39%] vs. 2 [6%], P = 0.002). Assessment of anxiety using GAD-7 and insomnia using ISI showed that PD patients tended to be more likely to present with clinical anxiety and insomnia, but neither reached significance (clinical anxiety among PD patients vs. controls: 19 (48%) vs. 11 (34%), P = 0.223; clinical insomnia among PD patients vs. controls: 13 (33%) vs. 7 (21%), P = 0.286) (Table 2). When analyzed by sex, significantly more female PD patients presented with clinical depression (PD patients vs. controls: 7 [50%] vs. 1 [3.8%], P = 0.001), whereas no significant difference was observed in male PD

**Table 1. Demographic characteristics of responders.**

| Characteristics | | No. (%) | | | |
| --- | --- | --- | --- | --- | --- |
| | | Total | PD | Control | P-value |
| Overall | | 71 (100) | 39 (54.9) | 32 (45.0) | |
| Sex | | | | | |
| | Male | 30 (42.2) | 25 (64.1) | 5 (15.6) | **< 0.001** |
| | Female | 41 (57.7) | 14 (35.8) | 27 (84.3) | |
| Age, y | | | | | |
| | <70 | 25 (35.2) | 11 (15.4) | 14 (19.7) | 0.078 |
| | 70–79 | 33 (46.4) | 18 (46.1) | 15 (46.8) | |
| | >80 | 13 (18.3) | 10 (25.6) | 3 (9.3) | |
| Living alone | | 3 (4.2) | 3 (7.6) | - | 0.109 |
| Disease duration, years | | | | | |
| | < 5 | | 22 (56.4) | - | - |
| | ≥ 5 | | 17 (43.5) | - | - |
| HY stage | | | | | |
| | stage 0–2 | | 12 (30.7) | - | - |
| | stage 3, 4 | | 27 (69.2) | - | - |
| Non-motor symptoms | | | | | |
| | Cognitive impairment | | 6 (15.3) | - | - |
| | Hallucinations | | 4 (10.2) | - | - |
| RBD | | | 6 (15.3) | - | - |
| L-DOPA, mg | | | | | |
| | < 600 | | 30 (76.9) | - | - |
| | ≥ 600 | | 9 (23.0) | - | - |
| Other medications | | | | | |
| | Dopamine agonist | | 15 (38.4) | - | - |
| | Psychiatric medicines | | 8 (20.5) | - | - |
| | Sleeping medicines | | 13 (33.3) | - | - |
| MDS-UPDRS part 2, median (IQR), n = 36* | | | 17 (10–20.5) | - | - |

Abbreviations: HY, Hoehn & Yahr; RBD, REM sleep behavior disorder; MDS-UPDRS, Movement Disorder Society Unified Parkinson's Disease Rating Scale; IQR, interquartile range.

*Of the 39 participants, 36 responded to the question about MDS-UPDRS part 2.

**Table 2. Severity categories of depression, anxiety, and insomnia measurements in PD patients and controls.**

| Severity category | | No. (%) | | | P-value* |
|---|---|---|---|---|---|
| | | Total | PD | Control | |
| PHQ-9, depression symptoms | | n = 69 | n = 38 | n = 31 | |
| | Normal | 32 (46.3) | 14 (36.8) | 18 (58.0) | **0.002** |
| | Mild | 20 (28.9) | 9 (23.6) | 11 (0.35) | |
| | Moderate | 9 (13.0) | 8 (21.0) | 1 (3.2) | |
| | Severe | 8 (11.5) | 7 (18.4) | 1 (3.2) | |
| GAD-7, anxiety symptoms | | n = 71 | n = 39 | n = 32 | |
| | Normal | 35 (49.2) | 18 (46.1) | 17 (53.1) | 0.223 |
| | Mild | 20 (28.1) | 9 (23.0) | 11 (34.3) | |
| | Moderate | 10 (14.0) | 7 (17.9) | 3 (9.3) | |
| | Severe | 6 (8.4) | 5 (12.8) | 1 (3.1) | |
| ISI, insomnia symptoms | | n = 71 | n = 39 | n = 32 | |
| | Absence | 31 (43.6) | 15 (38.4) | 16 (50.0) | 0.286 |
| | Subthreshold | 20 (28.1) | 11 (28.2) | 9 (28.1) | |
| | Moderate | 18 (25.3) | 11 (28.2) | 7 (21.8) | |
| | Severe | 2 (2.8) | 2 (5.1) | 0 (0.0) | |

Abbreviations: PHQ-9, 9-item Patient Health Questionnaire; GAD-7, 7-item Generalized Anxiety Disorder; ISI, 7-item Insomnia Severity Index.

* Cutoff scores for PHQ-9, GAD-7, and ISI were 10, 7, and 15, respectively. Participants who had scores greater than the cutoff threshold were characterized as showing clinical symptoms. Chi-squared test was applied to compare the rate of participants with clinical symptoms in PD patients and controls.

patients and controls (PD patients vs. controls: 8 [33%] vs. 1 [20%], P = 0.558) (S2A **and** S2**B Table**).

The median (IQR) score of PHQ-9 for depression in PD patients was significantly higher than in controls (PD patients vs. controls: 7.0 [2.0–13.0] vs. 2.0 [0.0–7.0], P = 0.010). Scores of GAD-7 for anxiety and ISI for insomnia tended to be higher in PD patients, but neither reached significance (median [IQR] GAD-7 scores among PD patients vs. controls: 6.0 [1.5–10.5] vs. 4.0 [1.0–7.0], P = 0.130; median [IQR] ISI scores among PD patients vs. controls: 10.0 [5.0–16.0] vs. 7.5 [2.0–13.25], P = 0.170) (**Table 3**). When analyzed by sex, no significant differences between PD patients and controls in depression, anxiety, and insomnia were observed for either sex (S3**A and** S3**B Table**).

**Table 3. Scores of depression, anxiety, and insomnia measurements in PD patients and controls.**

| Scale | Median (IQR) | | | P-value* |
|---|---|---|---|---|
| | Total score | PD | Control | |
| PHQ-9, depression symptoms | 5.0 | 7.0 | 2.0 | **0.010** |
| | (1.0–8.0) | (2.0–13.0) | (0.0–7.0) | |
| GAD-7, anxiety symptoms | 5.0 | 6.0 | 4.0 | 0.130 |
| | (1.0–9.0) | (1.5–10.5) | (1.0–7.0) | |
| ISI, Insomnia symptoms | 8.0 | 10.0 | 7.5 | 0.170 |
| | (2.5–15.0) | (5.0–16.0) | (2.0–13.25) | |

Abbreviations: PD, Parkinson's disease; IQR, interquartile range; PHQ-9, 9-item Patient Health Questionnaire; GAD-7, 7-item Generalized Anxiety Disorder; ISI, 7-item Insomnia Severity Index.

* Mann-Whitney U test was adapted to compare each questionnaire score between PD patients and controls on.

**Table 4. Rate of responders with worsening of motor performance, anxiety, and insomnia.**

|  |  | No. (%) |  |  |  |
|---|---|---|---|---|---|
| Values |  | Total | PD | Control | P-value |
| Overall |  | 71 (100) | 39 (54.9) | 32 (45.0) |  |
|  | Worsening of motor performance | 28 (39.4) | 16 (41.0) | 12 (37.5) | 0.84 |
|  | Worsening of anxiety | 26 (36.6) | 12 (30.7) | 14 (43.7) | 0.25 |
|  | Worsening of insomnia | 8 (11.2) | 5 (12.8) | 3 (9.3) | 0.64 |

Abbreviation: PD, Parkinson's disease.

Among both PD patients and controls, 30–40% of participants complained of subjective worsening of motor performance and anxiety, but there were no significant differences (subjective worsening of motor performance among PD patients vs. controls: 16 [41%] vs. 12 [37%], P = 0.84; subjective worsening of anxiety among PD patients vs. controls: 12 [30%] vs. 14 [43%], P = 0.25). Interestingly, the control group was more likely to report subjective worsening of anxiety than PD patients. Fewer PD patients and controls complained of subjective worsening of insomnia (PD patients vs. controls: 5 [12%] vs. 3 [9%], P = 0.64) (**Table 4**). When analyzed by sex, more than 40% of female participants in both PD and control groups complained of subjective worsening of motor function and anxiety, whereas less than 20% of male participants in PD and control groups showed subjective worsening of motor function and 20–40% of them showed subjective anxiety (S4**A and** S4**B Table**).

## Risk factors

Multivariate logistic regression analysis demonstrated that an MDS-UPDRS part 2 score was correlated with the presence of clinical depression (PHQ-9 score $\geq$ 10) and clinical anxiety (GAD-7 score $\geq$ 7) in PD patients (clinical depression: OR, 1.31; 95% CI, 1.04–1.66; P = 0.025; clinical anxiety: OR, 1.36; 95% CI, 1.07–1.72; P = 0.013). A male sex was also a significant risk factor for PD patients' clinical anxiety in the multivariate model (OR, 17.12; 95% CI, 1.13–257.27; P = 0.040). Univariate logistic regression analysis showed that an MDS-UPDRS part 2 score was correlated with clinical insomnia (clinical insomnia: OR, 1.13; 95% CI, 1.01–1.26; P = 0.038), but it did not reach significance in the multivariate model (**Table 5**). The univariate model also showed the following results: the L-DOPA dose was not associated with clinical depression, anxiety, or insomnia, and the use of dopamine agonists was significantly correlated with clinical depression (clinical depression: OR, 5.40; 95% CI, 1.29–22.60; P = 0.021), but not with clinical insomnia or anxiety. In the multivariate model, the use of dopamine agonists did not reached significance (**Table 6**). Neither uni- nor multivariate logistic regression analyses were able to demonstrate significant risk factors for subjective worsening of motor performance, anxiety, and insomnia (**Table 7**).

## Discussion

This cross-sectional study in Japan involved 39 PD patients and 32 controls and revealed that PD patients were significantly more likely to have clinical depression than controls under the social restrictions imposed due to the COVID-19 pandemic. An MDS-UPDRS part 2 score was correlated with the presence of clinical depression and anxiety in PD patients. A male sex was a significant risk factor only for clinical anxiety in PD patients. Subjective worsening of motor performance was noted in about 40% of both PD patients and controls. Subjective worsening of anxiety was more common in controls, though it did not reach statistical significance.

**Table 5. Risk factors for neuropsychiatric symptoms identified by uni- and multivariate logistic regression analyses.**

| Variable | | No. of clinical cases/No. of total cases (%) | Univariate model | | | | | | Multivariate model | | | | | |
|---|---|---|---|---|---|---|---|---|---|---|---|---|---|---|
| | | | Unadjusted OR | 95%CI | | | P-value | | Adjusted OR | 95%CI | | | P-value | |
| | | | | | | | Category | Overall | | | | | Category | Overall |
| PHQ-9, depression symptoms | | | | | | | | | | | | | | |
| Sex | | | | | | | | | | | | | | |
| | Female | 7/14 (50.0) | 1 [reference] | | | | - | | 1 [reference] | | | | - | |
| | Male | 8/24 (33.3) | 0.5 | (0.13 | - | 1.93) | 0.314 | | 5.66 | (0.51 | - | 62.47) | 0.157 | |
| Age | | | | | | | | | | | | | | |
| | <70 | 7/11 (63.6) | 1 [reference] | | | | - | 0.123 | 1 [reference] | | | | - | 0.530 |
| | 70–79 | 4/17 (23.5) | 0.18 | (0.03 | - | 0.93) | 0.084 | | 0.61 | (0.05 | - | 8.08) | 0.750 | |
| | ≥80 | 4/10 (40.0) | 0.38 | (0.07 | - | 2.22) | 0.901 | | 0.19 | (0.01 | - | 3.93) | 0.264 | |
| Disease duration | | | | | | | | | | | | | | |
| | < 5 | 5/22 (22.7) | 1 [reference] | | | | - | | 1 [reference] | | | | - | |
| | ≥ 5 | 10/16 (62.5) | 5.67 | (1.37 | - | 23.46) | 0.017 | | 1.01 | (0.13 | - | 7.78) | 0.995 | |
| HY stage | | | | | | | | | | | | | | |
| | 0–2 | 4/12 (33.3) | 1 [reference] | | | | - | | 1 [reference] | | | | - | |
| | 3, 4 | 11/26 (42.3) | 1.47 | (0.35 | - | 6.13) | 0.600 | | 10.17 | (0.57 | - | 182.91) | 0.116 | |
| MDS-UPDRS part 2* | | | 1.24 | (1.06 | - | 1.45) | 0.008 | | 1.31 | (1.04 | - | 1.66) | 0.025 | |
| GAD-7, anxiety symptoms | | | | | | | | | | | | | | |
| Sex | | | | | | | | | | | | | | |
| | Female | 7/14 (50.0) | 1 [reference] | | | | - | | 1 [reference] | | | | - | |
| | Male | 12/25 (48.0) | 0.92 | (0.25 | - | 3.42) | 0.905 | | 17.12 | (1.13 | - | 257.27) | 0.040 | |
| Age | | | | | | | | | | | | | | |
| | <70 | 8/11 (72.7) | 1 [reference] | | | | - | 0.137 | 1 [reference] | | | | - | 0.458 |
| | 70–79 | 6/18 (33.3) | 0.19 | (0.04 | - | 0.98) | 0.083 | | 0.55 | (0.04 | - | 7.69) | 0.774 | |
| | ≥80 | 5/10 (50.0) | 0.38 | (0.06 | - | 2.31) | 0.850 | | 0.16 | (0.01 | - | 3.08) | 0.221 | |
| Disease duration | | | | | | | | | | | | | | |
| | < 5 | 9/22 (40.9) | 1 [reference] | | | | - | | 1 [reference] | | | | - | |
| | ≥ 5 | 10/17 (58.8) | 2.06 | (0.57 | - | 7.47) | 0.270 | | 0.35 | (0.04 | - | 3.08) | 0.359 | |
| HY stage | | | | | | | | | | | | | | |
| | 0–2 | 6/12 (50.0) | 1 [reference] | | | | - | | 1 [reference] | | | | - | |
| | 3, 4 | 13/27 (48.1) | 0.93 | (0.24 | - | 3.62) | 0.915 | | 8.19 | (0.52 | - | 128.74) | 0.135 | |
| MDS-UPDRS part 2* | | | 1.17 | (1.03 | - | 1.32) | 0.014 | | 1.36 | (1.07 | - | 1.72) | 0.013 | |
| ISI, insomnia symptoms | | | | | | | | | | | | | | |
| Sex | | | | | | | | | | | | | | |
| | Female | 5/14 (35.7) | 1 [reference] | | | | - | | 1 [reference] | | | | - | |
| | Male | 8/25 (32.0) | 0.85 | (0.21 | - | 3.36) | 0.814 | | 1.68 | (0.25 | - | 11.2) | 0.595 | |
| Age | | | | | | | | | | | | | | |
| | <70 | 6/11 (54.6) | 1 [reference] | | | | - | 0.116 | 1 [reference] | | | | - | 0.310 |
| | 70–79 | 3/18 (16.7) | 0.17 | (0.03 | - | 0.93) | 0.052 | | 0.21 | (0.03 | - | 1.63) | 0.158 | |
| | ≥80 | 4/10 (40.0) | 0.56 | (0.1 | - | 3.15) | 0.693 | | 0.53 | (0.05 | - | 5.7) | 0.884 | |
| Disease duration | | | | | | | | | | | | | | |
| | < 5 | 6/22 (27.3) | 1 [reference] | | | | - | | 1 [reference] | | | | - | |
| | ≥ 5 | 7/17 (41.2) | 1.87 | (0.49 | - | 7.18) | 0.364 | | 1.55 | (0.27 | - | 8.78) | 0.620 | |
| HY stage | | | | | | | | | | | | | | |
| | 0–2 | 5/12 (41.7) | 1 [reference] | | | | - | | 1 [reference] | | | | - | |
| | 3, 4 | 8/27 (29.6) | 0.59 | (0.14 | - | 2.42) | 0.464 | | 0.87 | (0.11 | - | 7.09) | 0.900 | |
| MDS-UPDRS part 2* | | | 1.13 | (1.01 | - | 1.26) | 0.038 | | 1.11 | (0.98 | - | 1.25) | 0.113 | |

Abbreviations: Unadjusted OR, Unadjusted odds ratio; 95%CI, 95% confidence interval; PHQ-9, 9-item Patient Health Questionnaire; HY, Hoehn & Yahr; MDS-UPDRS, Movement Disorder Society-Unified Parkinson's Disease Rating Scale; GAD-7, 7-item Generalized Anxiety Disorder; ISI, 7-item Insomnia Severity Index.

* Continuous quantity was analyzed.

It is becoming clear that the COVID-19 pandemic has a negative impact on the PD symptoms [8, 9, 20]. Van der Heide et al. showed that COVID-19-related stress load was positively correlated with psychological distress, resulting in more severe PD symptoms [14]. Shalash et al. showed that more than 52% of PD patients reported stress, anxiety, and disrupted contact

**Table 6. Association with neuropsychiatric symptoms and PD drugs in uni- and multivariate logistic regression analyses.**

| Variable | | No. of clinical cases/No. of total cases (%) | Univariate model | | | | Multivariate model* | | | |
|---|---|---|---|---|---|---|---|---|---|---|
| | | | Unadjusted OR | 95%CI | | P-value | Adjusted OR | 95%CI | | P-value |
| **PHQ-9, depression symptoms** | | | | | | | | | | |
| L-DOPA, mg | | | | | | | | | | |
| | < 600 | 10/30 (33.3) | 1 [reference] | | | - | 1 [reference] | | | - |
| | ≥ 600 | 5/8 (62.5) | 3.33 | (0.66 - | 16.85) | 0.145 | 1.39 | (0.13 - | 15.26) | 0.787 |
| Dopamine agonist | | | | | | | | | | |
| | without | 6/24 (25.0) | 1 [reference] | | | - | 1 [reference] | | | - |
| | with | 9/14 (64.3) | 5.40 | (1.29 - | 22.6) | **0.021** | 9.33 | (0.85 - | 102.72) | 0.068 |
| **GAD-7, anxiety symptoms** | | | | | | | | | | |
| L-DOPA, mg | | | | | | | | | | |
| | < 600 | 15/30 (50.0) | 1 [reference] | | | - | | | | |
| | ≥ 600 | 4/9 (44.4) | 0.8 | (0.18 - | 3.57) | 0.77 | | | | |
| Dopamine agonist | | | | | | | | | | |
| | without | 9/24 (72.7) | 1 [reference] | | | - | 1 [reference] | | | - |
| | with | 10/15 (66.7) | 3.33 | (0.86 - | 12.92) | 0.082 | 13.07 | (0.81 - | 210.16) | 0.070 |
| **ISI, insomnia symptoms** | | | | | | | | | | |
| L-DOPA, mg | | | | | | | | | | |
| | < 600 | 9/30 (30.0) | 1 [reference] | | | - | | | | |
| | ≥ 600 | 4/9 (44.4) | 1.87 | (0.41 - | 8.61) | 0.424 | | | | |
| Dopamine agonist | | | | | | | | | | |
| | without | 8/24 (33.3) | 1 [reference] | | | - | | | | |
| | with | 5/15 (33.3) | 1.00 | (0.25 - | 3.93) | 1.000 | | | | |

Abbreviations: Unadjusted OR, Unadjusted odds ratio; 95%CI, 95% confidence interval; PHQ-9, 9-item Patient Health Questionnaire; GAD-7, 7-item Generalized Anxiety Disorder; ISI, 7-item Insomnia Severity Index.

with their physicians due to COVID-19 related social restrictions [9]. Basically, depression, anxiety, and sleep disturbance are common non-motor features from the prodromal to late stage of PD [4–6]. A previous systematic review of prevalence for depression in PD patients concluded that clinically significant depressive symptoms were observed in 35% of them [21]. Other studies using PHQ-9, like our study, for the assessment of depression reported the prevalence of clinical depression as 14–34% [22, 23]. Because we did not have data on the neuropsychiatric status of PD patients before the COVID-19 pandemic, we could not directly show that the COVID-19 pandemic in Japan worsened the neuropsychiatric status of our PD patients. However, the prevalence of clinical depression in PD patients of this study (39%) was as high as or higher than that of previous reviews. Considering this fact, the impact of the COVID-19 pandemic should not be ignored on the neuropsychiatric status of PD patients even in Japan, where no legal penalties were imposed for going out during the state of emergency.

PD patients with specific characteristics such as low optimism and high neuroticism were reported to show higher levels of psychological distress due to COVID-19 pandemic [14]. Previous reports described that PD patients with a maladaptive metacognitive style showed an increased vulnerability to psychological distress. In other words, uncontrolled anxiety in PD patients is likely to amplify the anxiety itself [24, 25]. In our study, clinical depression and anxiety were more common in PD patients, even though the prevalence of subjective worsening of anxiety was comparable between PD patients and controls. Overall, we may indicate that PD

**Table 7. Risk factors for subjective worsening of motor performance, anxiety, and insomnia identified by uni- and multivariate logistic regression analyses.**

| Variable | | No. of clinical cases/No. of total cases (%) | Univariate model | | | | | Multivariate model | | | | |
|---|---|---|---|---|---|---|---|---|---|---|---|---|
| | | | Unadjusted OR | 95%CI | | P-value Category | Overall | Adjusted OR | 95%CI | | P-value Category | Overall |
| **Subjective worsening of motor performance** | | | | | | | | | | | | |
| Sex | | | | | | | | | | | | |
| | Female | 8/14 (57.1) | 1 [reference] | | | - | | 1 [reference] | | | - | |
| | Male | 8/25 (32.0) | 0.35 | (0.09 - 1.36) | | 0.131 | | 0.60 | (0.11 - 3.20) | | 0.547 | |
| Age | | | | | | | | | | | | |
| | <70 | 6/11 (54.6) | 1 [reference] | | | - | 0.301 | 1 [reference] | | | - | 0.582 |
| | 70–79 | 5/18 (27.8) | 0.32 | (0.07 - 1.54) | | 0.126 | | 0.39 | (0.05 - 2.80) | | 0.307 | |
| | ≥80 | 5/10 (50.0) | 0.83 | (0.15 - 4.63) | | 0.606 | | 0.75 | (0.07 - 7.65) | | 0.845 | |
| Disease duration | | | | | | | | | | | | |
| | < 5 | 7/22 (31.8) | 1 [reference] | | | - | | 1 [reference] | | | - | |
| | ≥ 5 | 9/17 (52.9) | 2.41 | (0.65 - 8.92) | | 0.188 | | 1.90 | (0.38 - 9.61) | | 0.437 | |
| HY stage | | | | | | | | | | | | |
| | 0–2 | 4/12 (33.3) | 1 [reference] | | | - | | 1 [reference] | | | - | |
| | 3, 4 | 12/27 (44.4) | 1.60 | (0.39 - 6.62) | | 0.517 | | 3.68 | (0.39 - 34.3) | | 0.253 | |
| MDS-UPDRS part 2* | | | 1.06 | (0.97 - 1.15) | | 0.217* | | 1.02 | (0.93 - 1.12) | | 0.694* | |
| **Subjective worsening of anxiety** | | | | | | | | | | | | |
| Sex | | | | | | | | | | | | |
| | Female | 7/14 (50.0) | 1 [reference] | | | - | | 1 [reference] | | | - | |
| | Male | 5/25 (20.0) | 0.25 | (0.06 - 1.05) | | 0.058 | | 0.34 | (0.06 - 2.15) | | 0.253 | |
| Age | | | | | | | | | | | | |
| | <70 | 4/11 (36.4) | 1 [reference] | | | - | 0.561 | 1 [reference] | | | - | 0.721 |
| | 70–79 | 4/18 (22.2) | 0.50 | (0.09 - 2.62) | | 0.287 | | 1.18 | (0.12 - 12.04) | | 0.719 | |
| | ≥80 | 4/10 (40.0) | 1.17 | (0.20 - 6.80) | | 0.516 | | 2.61 | (0.16 - 42.16) | | 0.427 | |
| Disease duration | | | | | | | | | | | | |
| | < 5 | 5/22 (22.7) | 1 [reference] | | | - | | 1 [reference] | | | - | |
| | ≥ 5 | 7/17 (41.2) | 2.38 | (0.59 - 9.53) | | 0.221 | | 1.71 | (0.29 - 9.91) | | 0.551 | |
| HY stage | | | | | | | | | | | | |
| | 0–2 | 3/12 (25.0) | 1 [reference] | | | - | | 1 [reference] | | | - | |
| | 3, 4 | 9/27 (33.3) | 1.50 | (0.32 - 6.94) | | 0.604 | | 2.12 | (0.17 - 26.69) | | 0.561 | |
| MDS-UPDRS part 2* | | | 1.06 | (0.97 - 1.16) | | 0.218 | | 1.03 | (0.93 - 1.15) | | 0.552 | |
| **Subjective worsening of insomnia** | | | | | | | | | | | | |
| Sex | | | | | | | | | | | | |
| | Female | 3/14 (21.4) | 1 [reference] | | | - | | 1 [reference] | | | - | |
| | Male | 2/25 (8.0) | 0.32 | (0.05 - 2.19) | | 0.245 | | 0.705 | (0.07 - 7.52) | | 0.772 | |
| Age | | | | | | | | | | | | |

(*Continued*)

**Table 7.** (Continued)

| Variable | | No. of clinical cases/No. of total cases (%) | Univariate model | | | | | Multivariate model | | | | |
|---|---|---|---|---|---|---|---|---|---|---|---|---|
| | | | Unadjusted OR | 95%CI | | P-value Category | Overall | Adjusted OR | 95%CI | | P-value Category | Overall |
| | <70 | 3/11 (27.3) | 1 [reference] | | | - | 0.278 | 1 [reference] | | | - | 0.569 |
| | 70–79 | 1/18 (5.6) | 0.16 | (0.01 - | 1.75) | 0.302 | | 0.19 | (0.01 - | 4.16) | 0.412 | |
| | ≥80 | 1/10 (10.0) | 0.30 | (0.02 - | 3.45) | 0.812 | | 0.35 | (0.01 - | 8.82) | 0.874 | |
| Disease duration | | | | | | | | | | | | |
| | < 5 | 3/22 (13.6) | 1 [reference] | | | - | | 1 [reference] | | | - | |
| | ≥ 5 | 2/17 (11.8) | 0.84 | (0.12 - | 5.72) | 0.863 | | 0.66 | (0.05 - | 8.36) | 0.745 | |
| HY stage | | | | | | | | | | | | |
| | 0–2 | 2/12 (16.7) | 1 [reference] | | | - | | 1 [reference] | | | - | |
| | 3, 4 | 3/27 (11.1) | 0.63 | (0.09 - | 4.32) | 0.634 | | 1.90 | (0.08 - | 38.5) | 0.676 | |
| MDS-UPDRS part 2* | | | 1.09 | (0.97 - | 1.23) | 0.131 | | 1.11 | (0.96 - | 1.28) | 0.172 | |

Abbreviations: Unadjusted OR, Unadjusted odds ratio; 95%CI, 95% confidence interval; HY, Hoehn & Yahr; MDS-UPDRS, Movement Disorder Society-Unified Parkinson's Disease Rating Scale.

* Continuous quantity was analyzed.

patients, especially who have pre-existing psychological problems, are more likely to develop clinical depression in response to social distress.

In this study, a high MDS-UPDRS part 2 score was shown to be a risk factor for clinical depression and anxiety. This is consistent with previous studies suggesting that reduced exercise time was associated with worsening of both motor and non-motor symptoms under COVID-19 pandemic [12–14] and that motor fluctuation and impairment of activities of daily life were significant risk factors for the development of depression [26]. In addition, we showed that a male sex was associated with clinical anxiety, while a female sex was reported to be a potential risk factor for anxiety both in general population and in PD patients [26, 27]. In the previous report, Xia et al. showed lower scores of depression and anxiety in male PD patient than female, though it was not multivariate analysis [10]. In addition, we are currently under the distinct stress due to COVID-19 pandemic that did not exist in the past, so we need more cases to determine the impact of sex on the neuropsychiatric symptoms under COVID-19 pandemic. Other potential risk factors for non-motor symptoms such as age, disease duration, and HY stage did not show statistical significance in this study [4–6]. A larger sample size may be needed to examine the correlation between neuropsychiatric symptoms and these potential risk factors because they presented statistical significance in the univariate model (**Table 5**).

Reasons for the high prevalence of subjective worsening of anxiety in controls may be related to the fact that they were caregivers, in addition to the fact that 84% of them are female. Oppo et al. also reported that caregivers complained similar or higher levels of worsened mental stress than PD patients during home confinement due to COVID-19 pandemic (PD patients vs. caregivers, 43% vs 54%, respectively) [12]. It has been reported that caregivers of PD patients have a marked mental burden [28, 29]. Research based on interviews revealed that caregivers of PD patients complained of increased responsibility and insufficient time to take care of themselves [30, 31]. Regarding COVID-19, Lara et al. showed that 30% of patients with

Alzheimer's disease and 40% of their caregivers reported worsening mental health conditions due to the COVID-19 lockdown [32]. Oppo et al. also indicated that some non-motor symptoms such as mood, cognition, and urinary problems in PD patients caused additional mental burden on the caregivers [12]. In addition to caregivers' own health concerns, feeling responsible for PD patients' health status may have led to the high prevalence of subjective worsening of anxiety in controls in this study.

Limitations of this study include the absence of information about depression, anxiety, and insomnia of participants before COVID-19 pandemic and a female bias in sex ratio in control group. We can review patients' chart and find information about psychiatric or sleeping medicines to estimate pre-COVID-19 neuropsychiatric status of PD patients, though these are not perfect and would not be available for controls. As we mentioned above, several papers are now proving the negative impact of the COVID-19 pandemic on symptoms of PD patients, so we need to continue to monitor the impact of the COVID-19 pandemic on PD patients in Japan. Although we calculated the required number of samples to compare the prevalence of neuropsychiatric symptoms during COVID-19 pandemic in PD and control groups based on previous papers, we had a larger number of female participants in controls. We found more clinical depression among female PD patients, while there were no significant differences between male PD patients and controls. This result may be influenced by the small number of male controls. We may need to adjust the sex ratio of participants in future and need more participants to estimate risk factors which potentially influence neuropsychiatric symptoms of PD patients.

## Conclusions

PD patients may be more likely to develop clinical depression than those without PD in the presence of social stresses, such as a pandemic, even in Japan where no legal penalties were imposed during the state of emergency. Considering the significant correlation between high MDS-UPDRS part 2 scores and the complication of severe depression and anxiety in PD patients, such patients may require special attention regarding the development of neuropsychiatric symptoms not only during COVID-19 pandemic but also in the event of another major disaster in the future.

## Supporting information

**S1 Table. Demographic characteristics of responders stratified by sex.**
(DOCX)

**S2 Table.** a. Severity categories of depression, anxiety, and insomnia measurements in female PD patients and controls. b. Severity categories of depression, anxiety, and insomnia measurements in male PD patients and controls.
(DOCX)

**S3 Table.** a. Scores of depression, anxiety, and insomnia measurements in female PD patients and controls. b. Scores of depression, anxiety, and insomnia measurements in male PD patients and controls.
(DOCX)

**S4 Table.** a. Rate of female responders with worsening of motor performance, anxiety, and insomnia. b. Rate of male responders with worsening of motor performance, anxiety, and insomnia.
(DOCX)

**S1 Data. Questions that ask for subjective change in participants and original data of all participants.**
(XLSX)

## Author Contributions

**Conceptualization:** Fukiko Kitani-Morii, Takashi Kasai.

**Data curation:** Fukiko Kitani-Morii, Takuma Ohmichi, Makiko Shinomoto, Yuzo Fujino.

**Formal analysis:** Go Horiguchi, Satoshi Teramukai.

**Funding acquisition:** Takashi Kasai, Takuma Ohmichi.

**Project administration:** Fukiko Kitani-Morii, Takashi Kasai.

**Supervision:** Toshiki Mizuno.

**Writing – original draft:** Fukiko Kitani-Morii.

**Writing – review & editing:** Takashi Kasai, Go Horiguchi, Satoshi Teramukai, Takuma Ohmichi, Makiko Shinomoto, Yuzo Fujino, Toshiki Mizuno.

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
