## [Decision Letter · Decision Letter 0]

3 Dec 2020

PONE-D-20-34679

Risk factors for neuropsychiatric symptoms in patients with Parkinson’s disease during COVID-19 pandemic in Japan

PLOS ONE

Dear Dr. Kasai,

Thank you for submitting your manuscript to PLOS ONE. After careful consideration, we feel that it has merit but does not fully meet PLOS ONE’s publication criteria as it currently stands. Therefore, we invite you to submit a revised version of the manuscript that addresses the points raised during the review process.

We look forward to receiving your revised manuscript.

Kind regards,

Kensaku Kasuga

Academic Editor

PLOS ONE

Journal Requirements:

3. Please include additional information regarding the survey or questionnaire used in the study and ensure that you have provided sufficient details that others could replicate the analyses. For instance, if you developed a questionnaire as part of this study and it is not under a copyright more restrictive than CC-BY, please include a copy, in both the original language and English, as Supporting Information, or include a citation if it has been published previously.

4. Please provide further details on sample size and power calculations.

Additional Editor Comments (if provided):

This manuscript describes that PD patients were more likely to have depression than control, and showed an association between psychiatric symptoms, especially depression and anxiety, and ADL assessed with UPDRS part 2 in PD patients under the COVID-19 pandemic. Furthermore, this study showed caregiver’s stress as assessed by subjective worsening of anxiety.

This study would contribute to considerations for care of PD patients under stressful conditions such as the COVID-19 pandemic. As reviewer 1 mentioned, this paper should be published as soon as possible.

However, I think that a revision is needed to improve this manuscript.

As the authors noted in a limitation, PD patients were not compared to before the COVID-19 pandemic. Reviewer 2 has also pointed out this point (see reviewer’s comment). The prevalence of depression in PD patients in this study (39%) was actually comparable to a previous report (reference 19: 34%, 95%CI 27.9-40.6). It suggests that PD patients are likely to have depression even not under the COVID-19 pandemic. Alternatively, some PD patients could have less depression despite under stressful conditions. The authors should consider these points more with additional data or the previous reports.

Please respond to the reviewer’s and my comments to make the manuscript better.

Reviewers' comments:

Reviewer's Responses to Questions

**Comments to the Author**

1. Is the manuscript technically sound, and do the data support the conclusions?

Reviewer #1: Yes

Reviewer #2: No

2. Has the statistical analysis been performed appropriately and rigorously? 

Reviewer #1: Yes

Reviewer #2: Yes

3. Have the authors made all data underlying the findings in their manuscript fully available?

Reviewer #1: Yes

Reviewer #2: Yes

4. Is the manuscript presented in an intelligible fashion and written in standard English?

Reviewer #1: Yes

Reviewer #2: Yes

5. Review Comments to the Author

Reviewer #1: this study was well conducted.

this study was aimed to clarify the impact of social restrictions imposed due to the COVID-19 pandemic on neuropsychiatric symptoms in PD patients and to identify risk factors.

and they concluded, In the presence of social restrictions, more attention needs to be paid to the neuropsychiatric complications of PD patients, especially those with more severe motor symptoms.

this paper must be published urgently.

Reviewer #2: Although the authors performed statistical analyses rigorously, I think there is a critical flaw in the method of this research. If the authors wanted to assess impact of the COVID-19 pandemic on symptoms of PD, they should have compared those of pre- with those of post-COVID-19 pandemic, or at least tried to measure changes between pre- and post-COVID-19 pandemic in more detailed items and also in multipoint scale. It seems that only self-reported dichotomy data about changes in three symptoms (motor performance, anxiety and insomnia) were collected in this article. Thus, this study mainly assessed only association between the severity of motor symptoms and that of non-motor symptoms in PD patients regardless of the COVID-19 restrictions. It is important itself but it does not answer the research question which the authors raised. Also, questionnaire survey could be rather easily conducted in a multi-site study setting to collect more patients.

6. PLOS authors have the option to publish the peer review history of their article (what does this mean?). If published, this will include your full peer review and any attached files.

Reviewer #1: No

Reviewer #2: No

---

## [Author Response · Author response to Decision Letter 0]

10 Dec 2020

Dear Editors and Reviewers,

We deeply appreciate your informative suggestions. Here, we submit following documents: Manuscript (without track change), Revised manuscript with track changes, Response to reviewers, and revised Supporting information data. Thank you very much for your consideration.

Journal Requirements:

(Answer) We appreciate the editor’s suggestion. We have corrected the manuscript as these PDF indicating.

(Answer) We appreciate the editor’s comment. We have added the additional information about informed consent as follows (P1, Line 69-70): Written informed consent was provided by all survey participants. No minors were among the participants. 

3. Please include additional information regarding the survey or questionnaire used in the study and ensure that you have provided sufficient details that others could replicate the analyses. For instance, if you developed a questionnaire as part of this study and it is not under a copyright more restrictive than CC-BY, please include a copy, in both the original language and English, as Supporting Information, or include a citation if it has been published previously.

(Answer) We appreciate the editor’s comment. Validated Japanese questionnaires were introduced in ref. 15-18. We have added information about questions that ask for subjective mental/physical changes in participants to Supporting information data file as follows (P18, Line371 and Supporting information data file): Has the COVID-19 pandemic made your anxiety worse? Has the COVID-19 pandemic made your insomnia worse? Has the COVID-19 pandemic made your motor performance worse? (originally in Japanese)

4. Please provide further details on sample size and power calculations.

(Answer) We appreciate the editor’s comment. We have added the sentence as follows (P3, Line71-74): According to a priori power analysis (with 95% power and 5% type I error rate) of the results of previous studies, the minimum number for the sample was found to be 41 patients for the comparative study of the prevalence of psychiatric symptoms in PD and control groups.

-Additional Editor Comments (if provided):

-This manuscript describes that PD patients were more likely to have depression than control, and showed an association between psychiatric symptoms, especially depression and anxiety, and ADL assessed with UPDRS part 2 in PD patients under the COVID-19 pandemic. Furthermore, this study showed caregiver’s stress as assessed by subjective worsening of anxiety.

This study would contribute to considerations for care of PD patients under stressful conditions such as the COVID-19 pandemic. As reviewer 1 mentioned, this paper should be published as soon as possible.

However, I think that a revision is needed to improve this manuscript.

As the authors noted in a limitation, PD patients were not compared to before the COVID-19 pandemic. Reviewer 2 has also pointed out this point (see reviewer’s comment). The prevalence of depression in PD patients in this study (39%) was actually comparable to a previous report (reference 19: 34%, 95%CI 27.9-40.6). It suggests that PD patients are likely to have depression even not under the COVID-19 pandemic.

(Answer) We deeply appreciate the editor’s informative suggestion. As the editor and reviewer pointed out, we did not show the information about pre-COVID-19 status. However, recent accumulating articles are proving that the COVID-19 pandemic had a negative impact on the PD symptoms (Van der Heide et al. J Parkinson Dis. 2020, Subramanian et al. npj Parkinsons Dis. 2020, Song et al. Parkinsonism Relat Disord. 2020). Even though Japanese state of emergency impose no legal penalties unlike in European countries, we cannot ignore the impact of COVID-19 pandemic on neuropsychiatric symptoms in Japanese PD patients. Based on the editor and reviewer’s comment, we have corrected the manuscript as follows (P13, Line196-210): It is becoming clear that the COVID-19 pandemic has a negative impact on the PD symptoms (8, 9, 20). Van der Heide et al. showed that COVID-19-related stress load was positively correlated with psychological distress, resulting in more severe PD symptoms (14). Shalash et al. showed that more than 52% of PD patients reported stress, anxiety, and disrupted contact with their physicians due to COVID-19 related social restrictions (9). Basically, depression, anxiety, and sleep disturbance are common non-motor features from the prodromal to late stage of PD (4-6). A previous systematic review of prevalence for depression in PD patients concluded that clinically significant depressive symptoms were observed in 35% of them (21). Other studies using PHQ-9, like our study, for the assessment of depression reported the prevalence of clinical depression as 14-34% (22, 23). Because we did not have data on the neuropsychiatric status of PD patients before the COVID-19 pandemic, we could not directly show that the COVID-19 pandemic in Japan worsened the neuropsychiatric status of our PD patients. However, the prevalence of clinical depression in PD patients of this study (39%) was as high as or higher than that of previous reviews. Considering this fact, the impact of the COVID-19 pandemic should not be ignored on the neuropsychiatric status of PD patients even in Japan, where no legal penalties were imposed for going out during the state of emergency.

-Alternatively, some PD patients could have less depression despite under stressful conditions. The authors should consider these points more with additional data or the previous reports.

Please respond to the reviewer’s and my comments to make the manuscript better.

(Answer) We appreciate the editor’s comment. As the editor mentioned, pre-existing specific characteristics (such as high neuroticism and low optimism) may be potential risk factors for worsening of psychological problems. We mentioned this point in Line211-212.

-Comments to the Author 

-Reviewer #1: this study was well conducted.

-this study was aimed to clarify the impact of social restrictions imposed due to the COVID-19 pandemic on neuropsychiatric symptoms in PD patients and to identify risk factors.

and they concluded, In the presence of social restrictions, more attention needs to be paid to the neuropsychiatric complications of PD patients, especially those with more severe motor symptoms.

this paper must be published urgently.

-Reviewer #2: Although the authors performed statistical analyses rigorously, I think there is a critical flaw in the method of this research. If the authors wanted to assess impact of the COVID-19 pandemic on symptoms of PD, they should have compared those of pre- with those of post-COVID-19 pandemic, or at least tried to measure changes between pre- and post-COVID-19 pandemic in more detailed items and also in multipoint scale. It seems that only self-reported dichotomy data about changes in three symptoms (motor performance, anxiety and insomnia) were collected in this article. Thus, this study mainly assessed only association between the severity of motor symptoms and that of non-motor symptoms in PD patients regardless of the COVID-19 restrictions. It is important itself but it does not answer the research question which the authors raised.

(Answer) We greatly appreciate the reviewer’s informative comments. As the editor and reviewer pointed out, we did not show the information about pre-COVID-19 status. However, recent accumulating articles are proving that the COVID-19 pandemic had a negative impact on the PD symptoms (Van der Heide et al. J Parkinson Dis. 2020, Subramanian et al. npj Parkinsons Dis. 2020, Song et al. Parkinsonism Relat Disord. 2020). Even though Japanese state of emergency impose no legal penalties unlike in European countries, we cannot ignore the impact of COVID-19 pandemic on neuropsychiatric symptoms in Japanese PD patients. Based on the editor and reviewer’s comment, we have corrected the manuscript as follows (P13, Line196-210): It is becoming clear that the COVID-19 pandemic has a negative impact on the PD symptoms (8, 9, 20). Van der Heide et al. showed that COVID-19-related stress load was positively correlated with psychological distress, resulting in more severe PD symptoms (14). Shalash et al. showed that more than 52% of PD patients reported stress, anxiety, and disrupted contact with their physicians due to COVID-19 related social restrictions (9). Basically, depression, anxiety, and sleep disturbance are common non-motor features from the prodromal to late stage of PD (4-6). A previous systematic review of prevalence for depression in PD patients concluded that clinically significant depressive symptoms were observed in 35% of them (21). Other studies using PHQ-9, like our study, for the assessment of depression reported the prevalence of clinical depression as 14-34% (22, 23). Because we did not have data on the neuropsychiatric status of PD patients before the COVID-19 pandemic, we could not directly show that the COVID-19 pandemic in Japan worsened the neuropsychiatric status of our PD patients. However, the prevalence of clinical depression in PD patients of this study (39%) was as high as or higher than that of previous reviews. Considering this fact, the impact of the COVID-19 pandemic should not be ignored on the neuropsychiatric status of PD patients even in Japan, where no legal penalties were imposed for going out during the state of emergency.

-As the reviewer pointed out, we need more detailed items and multipoint scale to assess subtle changes in symptoms in PD patients due to COVID-19 pandemic. The important previous paper (Van der Heide et al. J Parkinsons Dis. 2020) used 9-point scale to assess PD symptoms, however, they eventually analyzed the data into three groups: worse, no change, and improved. In addition, they showed that non-responders had a higher severity of PD symptoms. Given that our response rates was as high as 88% in PD patients, we believe that the simplified assessment method had some effect in avoiding respondent bias.

Also, questionnaire survey could be rather easily conducted in a multi-site study setting to collect more patients.

(Answer) We appreciate the reviewer’s important suggestion. Sample size calculation based on previous papers indicated that the number of participants needed to compare the prevalence of neuropsychiatric symptoms in PD and control group was 41 for each group, but we need more participants to assess risk factors by multivariate analysis. However, changes in work schedules of physicians at each facility due to stay-at-home order made multi-site study difficult. Based on the above, we have corrected the manuscript as follows (P14, Line 252-269): Although we calculated the required number of samples to compare the prevalence of neuropsychiatric symptoms during COVID-19 pandemic in PD and control groups based on previous papers, we had a larger number of female participants in controls. We found more clinical depression among female PD patients, while there were no significant differences between male PD patients and controls. This result may be influenced by the small number of male controls. We may need to adjust the sex ratio of participants in future and need more participants to estimate risk factors which potentially influence neuropsychiatric symptoms of PD patients.

---

## [Decision Letter · Decision Letter 1]

11 Jan 2021

Risk factors for neuropsychiatric symptoms in patients with Parkinson’s disease during COVID-19 pandemic in Japan

PONE-D-20-34679R1

Dear Dr. Kasai,

We’re pleased to inform you that your manuscript has been judged scientifically suitable for publication and will be formally accepted for publication once it meets all outstanding technical requirements.

Kind regards,

Kensaku Kasuga

Academic Editor

PLOS ONE

Additional Editor Comments (optional):

The authors fully responded to the comments.

Reviewers' comments:

Reviewer's Responses to Questions

**Comments to the Author**

1. If the authors have adequately addressed your comments raised in a previous round of review and you feel that this manuscript is now acceptable for publication, you may indicate that here to bypass the “Comments to the Author” section, enter your conflict of interest statement in the “Confidential to Editor” section, and submit your "Accept" recommendation.

Reviewer #1: All comments have been addressed

Reviewer #2: All comments have been addressed

2. Is the manuscript technically sound, and do the data support the conclusions?

Reviewer #1: Yes

Reviewer #2: Yes

3. Has the statistical analysis been performed appropriately and rigorously? 

Reviewer #1: Yes

Reviewer #2: Yes

4. Have the authors made all data underlying the findings in their manuscript fully available?

Reviewer #1: Yes

Reviewer #2: Yes

5. Is the manuscript presented in an intelligible fashion and written in standard English?

Reviewer #1: Yes

Reviewer #2: Yes

6. Review Comments to the Author

Reviewer #1: This is well conducted study.

Reviewer #2: I still think the study design the authors employed is not the best for answering the research question they raised, however, they tried to cover this defect with reviewing recent literature in the article. Given that there are country/region -specific characteristics of patients with PD, a similar study from a different country/region is worth publishing and it should occur timely during the pandemic as the other reviewer wrote.

7. PLOS authors have the option to publish the peer review history of their article (what does this mean?). If published, this will include your full peer review and any attached files.

Reviewer #1: No

Reviewer #2: No

---

## [Editor Report · Acceptance letter]

13 Jan 2021

PONE-D-20-34679R1 

Risk factors for neuropsychiatric symptoms in patients with Parkinson’s disease during COVID-19 pandemic in Japan 

Dear Dr. Kasai:

I'm pleased to inform you that your manuscript has been deemed suitable for publication in PLOS ONE. Congratulations! Your manuscript is now with our production department. 

Kind regards, 

on behalf of

Dr. Kensaku Kasuga 

Academic Editor

PLOS ONE